# Technical note: incorporating expert domain knowledge into causal structure discovery workflows

Jarmo Mäkelä[1], Laila Melkas[1], Ivan Mammarella[2], Tuomo Nieminen[2,3], Suyog Chandramouli[1], Rafael Savvides[1], and Kai Puolamäki[1,2]

[1]Department of Computer Science, P.O. Box 68, FI-00014 University of Helsinki, Helsinki, Finland
[2]Institute for Atmospheric and Earth System Research / Physics, P.O. Box 64, FI-00014 University of Helsinki, Helsinki, Finland
[3]Institute for Atmospheric and Earth System Research / Forest Sciences, P.O. Box 27, FI-00014 University of Helsinki, Helsinki, Finland

**Correspondence:** Jarmo Mäkelä (jarmo.makela@helsinki.fi)

**Abstract.** In this note, we argue that the outputs of causal discovery algorithms should not usually be considered as end results but starting points and hypothesis for further study. The incentive to explore this topic came from a recent study by Krich et al. (2020), which gives a good introduction to estimating causal networks in biosphere–atmosphere interaction, but confines the scope by investigating the outcome of a single algorithm. We aim to give a broader perspective to this study and to illustrate how not only different algorithms, but also different initial states and prior information of possible causal model structures, affect the outcome. We provide a proof-of-concept demonstration of how to incorporate expert domain knowledge with causal structure discovery and remark on how to detect and take into account over-fitting and concept drift.

## 1   Main text

In a recent paper Krich et al. (2020) tested and applied a newly developed PCMCI algorithm (Runge, 2020; Runge et al., 2019b) in order to detect causal links in geophysical data. PCMCI combines momentary conditional independence (MCI) tests with a standard constraint-based causal structure discovery (CSD) algorithm called PC, named after its authors Peter Spirtes and Clark Glymour. The PCMCI algorithm is used on flux tower eddy covariance data and related meteorological measurements of six variables in order to detect which variables can be seen to steer the behaviour of others. The paper can be viewed as a proof-of-concept and is a good introduction to causality and underlying problems, given the novelty of applying these types of methods to better understand biosphere-atmosphere interactions. However, we feel that the approach in Krich et al. (2020) – together with much of the other related work (Runge et al., 2019a) – is limited in its contribution to the practical application of CSD algorithms. There were items that in our opinion are significant that were only briefly mentioned or not at all addressed in Krich et al. (2020). These are:

- Different CSD algorithms may produce distinct outcomes (models) when operating on the same data. It is often difficult to identify the "correct" among these models, purely based on data.

- The choice of initial state (known structures) affects the behaviour and output of CSD algorithms. Due to their setup, Krich et al. (2020) employed an empty graph, but other choices are also possible.
- Utilising the knowledge of the domain experts and user interaction can be used to improve the models.
- Over-fitting and concept drift were addressed in (Krich et al., 2020) via the use of Akaike information criteria (AIC) (Akaike, 1974) but as these issues are central to any model selection or development we want to stress their importance. Over-fitting means that the analysis relies too much on the training data. Usually this happens when the amount of data is too small, resulting the causal model fitting to noise. Concept drift means that the underlying data distribution changes, rendering the causal model obsolete. An example of a concept drift is that a model trained on a certain location may not describe relations in another location; it is important to be able to take this phenomenon into account.

These comments are based on our recent workshop paper in the KDD 2021 conference (Melkas et al., 2021). Since many experts in Earth system sciences are not likely to follow said conference, we wanted to convey the main findings via this brief. In short, we try to find a model (directed acyclic graph) that best reflects the data, domain knowledge and user beliefs. Here we explore the behaviour of several CSD algorithms on both synthetic and real data and demonstrate how to incorporate prior knowledge and user interactions to this process. Before examining these topics in more detail, we present the underlying workflow in our approach:

1. Input domain knowledge (if any) as probabilities of known structures in the data.
2. Apply CSD algorithms to the data with the domain knowledge.
3. Choose a model from those provided by the algorithms, e.g. what the user regards as the best model in terms of their background knowledge and model score, which is a (user-defined) measure on how well the model fits the data.
4. Apply user interactions to the chosen model. We have substituted an actual user with a greedy search algorithm that examines the neighbouring models (one edit away) of the current one and chooses the best, in terms of model score.
5. Check the validity of the chosen model. We use cross-validation to detect over-fitting and concept drift due to its simplicity but other methods, e.g., AIC are possible as well.

The presented approach is Bayesian in nature and can be formulated as building a probabilistic model of the data. The aim is to find (locally) optimal model and as such, we assume that the domain knowledge can be characterised by a prior distribution over all possible causal structures (known features in the graph and confidence in that knowledge). Similarly, in our simulation, the user will have confidence, represented by parameter $k$, in certain structures between any pair of variables (A $\rightarrow$ B, B $\rightarrow$ A or no link). The user (in our case greedy search) is presented with options for simple edits (edge modifications) and how these edits would affect the model score. This process is iterated, until the current model is at least as good as any of the neighbours – see Melkas et al. (2021) for details. The outcomes are also compared to a model produced by actual domain experts (IM and TN). The takeaway message is that instead of using domain knowledge to merely quality check the final model produced by a CSD algorithm, the prior knowledge should be incorporated into the causal structure discovery process. The CSD methods we have used are constraint-based PC-stable (Colombo and Maathuis, 2014), score-based Greedy Equivalence Search (GES) (Chickering, 2003) and Linear Non-Gaussian Acyclic Model (LiNGAM) (Shimizu et al., 2006).

Constraint-based approaches use the idea that two statistically independent variables cannot be causally linked – for PC-stable we use statistical tests of conditional independence of structures with two significance levels: 0.1 and 0.01. PC-stable is a modified version of the PC algorithm with edits that reduces the dependency of the output to the order in which the variables are given. GES starts with an empty graph (equivalence class with no dependencies) and operates on two phases: it first iteratively adds simple structures to maximize the model score, and then iteratively considers all possible edge removals.

LiNGAM is based on independent component analysis (ICA) and exploits asymmetry, a fundamental property of causality (the relationship between cause and effect is asymmetrical).

We use both synthetic data as well as flux tower eddy covariance variables – same variables as in Krich et al. (2020) – measured at the SMEAR II station at Hyytiälä, Finland (Mammarella, 2020). All presented numerical analyses use synthetic data, which enables us to know the "true model". This data is created by generating a random (directed acyclic) graph and

sampling it with random edge weights to produce data sets of varying size. Each graph is generated with a sparsity of 0.3, which means that each pair of variables has an edge between them with a probability of 0.3. All edges are oriented away from the first variable and in the same order the variables are defined, which ensures acyclicity. Noise from either uniform distribution $U(-0.01, 0.01)$ or Gaussian with a standard deviation of 0.01 was added for each variable (for each variable the choice of the distribution was random). The reason for including both types of noise distributions is to create data sets which almost follow

assumptions made by the algorithms while still breaking some of them. All of the algorithms we use in the experiments assume linearity but, additionally, PC-Stable and GES assume Gaussianity of noise and LiNGAM assumes non-Gaussianity.

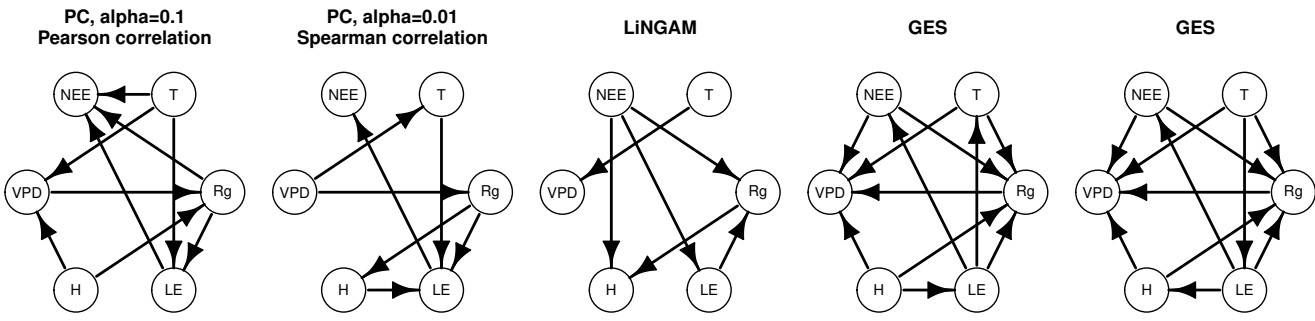

**Figure 1.** Different algorithms produce different causal graphs for the same data. PC algorithm is started from a full graph, LiNGAM has no defined initial graph and we started GES from an empty graph. GES produces (multiple) graphs with indistinguishable conditional dependency relationships.

## 2  Differences in CSD algorithms

While Krich et al. (2020) have focused on PCMCI, it is worthwhile to note that different CSD algorithms have varied outputs (models) for the same input data (Colombo and Maathuis, 2014; Druzdzel, 2009) since each algorithm operates differently and

makes different assumptions about the underlying data (Fig. 1). Additionally, even if the modelling assumptions in the causal

discovery process are correct, insufficient or biased data may result in skewed results. Therefore, the model gained from any one of these algorithms should not be viewed as the end result, but rather a starting point for further analysis. Often it is not clear, which among the discovered models is the "best", although we can argue that some of them are more plausible (Runge et al., 2019b), given the expert's knowledge. In some algorithms, inputting this prior knowledge (e.g., probabilities of certain

structures) is possible, but the ability to iteratively refine this background knowledge during the data analysis process nor the possibility to express uncertainty in the prior information have not been built in. These caveats hinder the usability of many CSD algorithms.

## 3   The choice of initial state

As different algorithms produce different models, so does the choice of initial state affect the outcome. These states can be, for

example, empty graphs, states produced by sampling methods, or states that reflect certain domain knowledge. Depending on the choice of initial state and on how uncertain the prior information is, different locally optimal models that fit the data may be found. Intuitively, it would be interesting to have a set of initial states that would cover all local optima, which could give rise to a global maximum-a-posteriori (MAP) solution. The underlying problem here would be to find a representative set of starting points for the exploration.

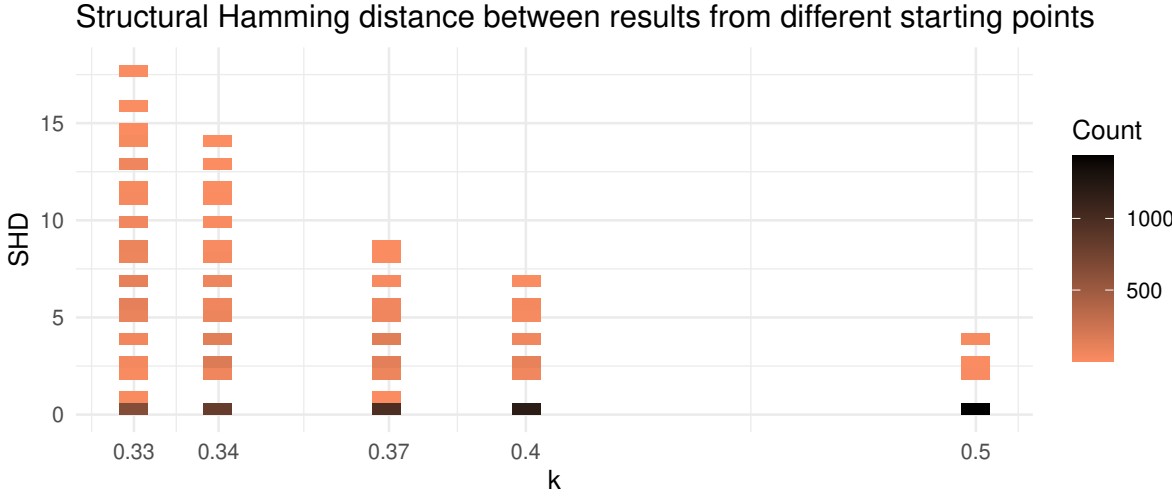

**Figure 2.** Pairwise structural Hamming distances when running analysis on the same data starting from different initial models. Variance in the distances show that the final model is affected by choice of initial model. Additionally, the spread of distances decreases rapidly with increasing prior knowledge.

We demonstrate the combined effect of utilising multiple initial states and different levels of prior knowledge ($k$) with synthetic data (Fig. 2). The initial states are generated by four different CSD algorithms and are complemented by an empty graph and the correct model, which we know as the data is synthetic. There are three possible causal states for a pair of model

variables $A$ and $B$: $A \rightarrow B$, $A \leftarrow B$, and no causal connection between the two. The user knows the correct state between each pair of variables with the probability of $k$, e.g. $k = 1/3$ means flat prior and $k = 1/2$ means that user knows the true states of the pairs with a probability of $1/2$. In these simulations, the level of prior knowledge $k \in [1/3, 1/2]$ We do not take into account wrong information ($k < 1/3$), and values above $1/2$ do not produce interesting results as such high certainty leads to near-constant results.

The structural Hamming distance (SHD) between two models indicates the minimum number of edge modifications (simple edits) required to transform either of the models into the other one. Even with a small amount of prior information, the end result after user interactions (greedy search) becomes much more stable – the spread of SHD diminishes as $k$ increases (Fig. 2).

## 4  Utilising domain knowledge and user interactions

The knowledge of the domain experts is classically used to provide suitable initial states for the CSD algorithms or to quality check the outcomes, but this knowledge should also be used to steer the CSD processes via user interactions and to allow reassessment of both user's own prior knowledge and related uncertainty as well as the algorithm process. When this knowledge is disregarded and the data is blindly trusted, any CSD algorithm or user (e.g., our greedy search) can uncover erroneous connections and miss relevant ones (Fig. 3). For example, the expert model (d) identifies four direct and well-established causal links from downwelling shortwave radiation (Rg) to latent and sensible heat fluxes (LE,H), temperature (T) and net ecosystem exchange (NEE). Two of these links (T and NEE) are missing from the best scoring model among the CSD algorithms (a), which also erroneously asserts that H is a driving force behind Rg. Both user models (b,c) find a new unrealistic link from Rg to vapour pressure deficit (VPD) and indicate that Rg is affecting T only indirectly through NEE.

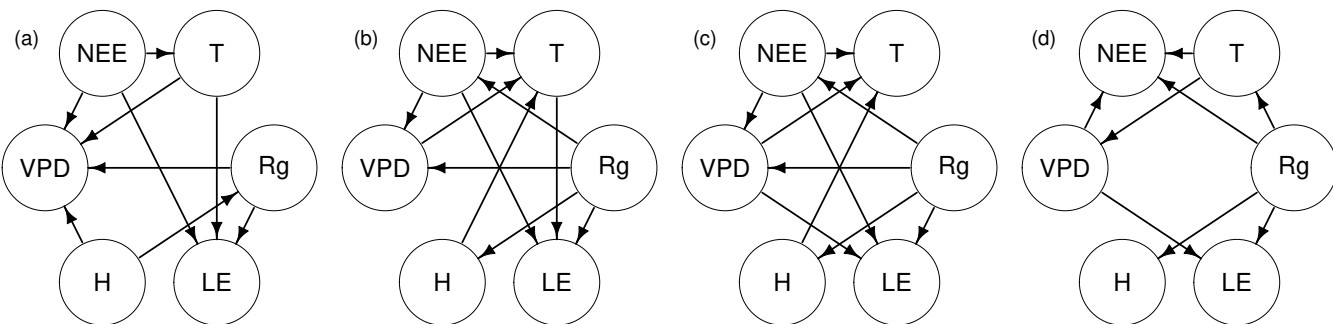

**Figure 3.** PC-stable produced the best scoring model (a) among our CSD algorithms, which was then further modified (b) by the greedy search. Greedy search produces a slightly different result (c) when initialised from an empty graph. The underlying causal structures were given a uniform prior ($k = 1/3$). Also shown is the expert model (d), produced by domain experts. The SHD from the expert model to (a),(b) and (c) are ten, seven and five.

## 5 Concluding remarks

Novel CSD algorithms, and more generally many machine learning methods, offer new insights in Earth system sciences. We argue that combining these methods with already abundant knowledge of the domain experts may yield more robust results and provide promising questions for future research. We also argue that while there are plethora of CSD algorithms that has been applied in earth sciences the question of how to use them in practice is still open. We have briefly presented here a fairly simple proof-of-concept approach as how to achieve this, demonstrated its effectiveness and highlighted some pitfalls – we direct anyone interested in a more detailed presentation to see Melkas et al. (2021), where we have also demonstrated how to detect over-fitting and concept drift, two common problems in statistical modelling, using $k$-fold blocked cross-validation (Bergmeir and Benítez, 2012). Hopefully, the work presented here will encourage developers to implement and study further interactive workflows.

*Author contributions.* JM prepared the comment, while LM ran the simulations and prepared the KDD manuscript under supervision of KP. IM and TN provided the domain expert knowledge and the expert model and together with SC and RS participated in producing this paper.

*Competing interests.* The authors declare that they have no competing of interests.

*Acknowledgements.* We thank Helsinki Institute for Information Technology, Future Makers Funding Program, and Finnish Center for Artificial Intelligence for support.

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
