# Peer review of "Comment on "Estimating causal networks in biosphere—atmosphere interaction with the PCMCI approach""

_Biogeosciences, 2021_

## Author Comment (AC1)

This document contains the referee comments to manuscript bg-2021-231: 'Comment on "Estimating causal networks in biosphere–atmosphere interaction with the PCMCI approach" and author responses to these comments. The comments are presented as indented *cursive* text and we have numbered them when needed.

**Referee 1 Comments**

The comment by Mäkelä et al. on the paper "Estimating causal networks in biosphere– atmosphere interaction with the PCMCI approach" by Krich et al. (including the reviewer) makes the point that the study should not take the outcome of a single causal discovery algorithm (here PCMCI) as an end result, but as a starting point and hypothesis for further study. They further illustrate on synthetic data how different prior expert knowledge affects such algorithms. The authors link to their recent workshop paper in the KDD 2021 conference (Melkas et al., 2021) which expands on the topic of "interactive" structure discovery.

1. Overall I deem this as a good and valid general point for any causal discovery analysis. However, I am not sure the commenting section is appropriate for this type of content since it does not specifically conduct an in-depth analysis of the paper to elaborate on how results would differ, but here it seems to mainly serve to advertise their workshop paper.

We do agree that the chosen approach is not ideal and we have explained our motivation in the manuscript introduction. Our main interest is to highlight some pitfalls and caveats related to causal structure discovery (CSD) algorithms and to present these in a more general manner than in the workshop paper. Ideally, we would not have to be referring to our workshop paper, but since there is very little other work in the nascent interactive CSD field available to illustrate our point, we refer the reader back to some of our work (which was motivated by wanting to avoid the exact pitfall that we point out here).

2. The authors present three different aspects of what they call "user interaction": (1) starting from a user-guided initial state, (2) expert-interactions during the execution of the causal discovery algorithm, and (3) overfitting and concept drift. These three points are discussed with very minimal examples and a few bits are unclear from the text: Are all numerical analyses conducted with synthetic data? What's the setup? Can point (2) be elaborated on a bit, it is hard to understand how this interaction is meant here.

There definitely seems to be a need to improve the manuscript on these points and to make efforts to expand and to better explain the setup. In the manuscript, we generally use different CSD algorithms to produce our initial states (or empty graph) – this would provide a real user multiple starting points to choose from e.g. what they regard as the best model in terms of model score and their background knowledge. In contrast, we provide the simulated user's background knowledge as likelihoods of graph structures (priors) and vary how much of this structure the "user" is aware of. The user (greedy search algorithm) then edits the graphs (starting from different initial states) using this knowledge. All presented numerical analyses use synthetic data, which enables us to know the "true model".

3. To put these comments in context with the actual paper: Yes, in the scope of this paper (Krich et al.) no initial prior knowledge (other than the choice of variables and that the type of dependency is linear) was used. However, the resulting graph was discussed from an expert perspective. The problem of overfitting was addressed in so far that the hyper-parameter (pc\_alpha) was optimized based on the Akaike Information Criterion, which is asymptotically equivalent to the cross-validation suggested in the comment. Indeed the paper can be viewed as a proof-of-concept and introduction to causality and underlying problems.

Essentially, we agree with everything presented in the 3rd comment. We felt that it was important to demonstrate both these concepts to the wider audience and underline that measures should be taken to avoid them. It is true that AIC is asymptotically equivalent to leave-one-out cross-validation (CV). However, we chose k-fold CV as our approach since it just looks at the test-set performance whereas AIC assumes that more parameters lead to a higher risk overfitting (and thus penalizes likelihood by the degrees of freedom). There are arguments both for and against each method and our deciding factor in favouring CV was simplicity. As a side note there is a nice comparison paper about these methods by Vehtari et al. (2016), the reference is at the end of this document.

4. As a remark, including expert knowledge into causal discovery is an interesting and not quite trivial problem. For example, while it may be easy to code-up (PCMCI's software package Tigramite has an option to start from a user-given initial graph), the completeness ("maximal informativeness") of causal discovery algorithms under expert knowledge is an open problem, at least for more complex scenarios such as the presence of hidden variables.

In the manuscript, we have not considered hidden or latent variables and we do agree with the comment. However, iterative user interaction still has a place as it may increase the user's understanding of the data and help in formulating beliefs about the potential data generating mechanisms. Additionally, more advanced methods could be developed to better identify and handle such situations.

**Referee 2 comments**

The paper raises a warning with regard to the blind usage of causal structure discovery (CSD) algorithms and correctly suggests that CSD should be seen more as a useful guidance in the understanding and modeling of multivariate systems rather than as the final outcome of analysis. In particular, the authors identify four main weaknesses of CSD: different causality methods produce different results, the outcome depends on the initial graph, domain knowledge is not taken into account, overfitting and performance drops due to distribution shifts are usually ignored. All these concerns are of great interest and yet have not been sufficiently explored in the literature. For this reason, the paper could have an impact on how CSD are deployed for scientific discovery. However, the authors do not satisfactorily develop the ideas presented in the introduction or at least do not provide enough details. Even for a first contribution towards the interesting interplay between domain knowledge and CSD I think that the authors should make an extra effort and expand further the content of the paper. Find below more detailed comments per section.

1. Main text: what do you mean by "outcomes (models) of causal structure discovery (CSD) algorithms are, in many cases, interchangeable"? are you referring to the fact that different CSD may produce distinct causal graph over the same data? Perhaps explain or rephrase this sentence. How exactly does a greedy search over models help injecting previous knowledge into CSD?

Agreed, the sentence needs to be rephrased. We are also referring to a situation when we have acquired multiple graphs and then should choose the "right" one. The greedy search itself should not be the focus here – it is our model of user behaviour. The greedy search is made aware of (some) of the expert knowledge and is left to modify the initial graph (be it from a CSD algorithm or empty graph) by searching the neighbouring states (one edit away) and choosing the new state based on model score.

2. Differences in CSD algorithms: the authors should provide a concrete example of a case in which different CSD produce different results (maybe taken from the literature). In its current form, section 2 is too generic and does not really add much information besides what is already stated in the introduction.

This type of image is readily available in Melkas et al. (2021) and we have added it here as an example (PC algorithm is usually started from a full graph, LiNGAM has no defined initial graph that is modified and we started GES from an empty graph; LiNGAM returns a single graph whereas both GES and PC return a Markov equivalence class, which can imply the presence of multiple graphs). Section 2 of the manuscript can likewise be expanded (it was left quite "light" as we assumed this would be generally known).

3. The choice of initial state: this is one of the main parts of the paper where the authors describe their first experiment on how the initial graph affects the result. Interestingly, they show that increasing the level of prior knowledge the outcomes gradually converge. However, more details on how the experiment is performed should be given. What are the synthetic data used? Which are the four CSD methods? How is the parameter "k" (encoding the prior) used to generate initial states?

This information can be added. The synthetic data description below is taken directly from Melkas et al (2021) but can be reworked into the manuscript. The text also mentions the used CSD methods: PC-Stable with two significance levels 0.1 and 0.01, GES, and ICA-based LiNGAM. Other algorithms were also considered but left out due to various reasons (too long runtime or in the case of FCI, comparability issues). The parameter "k" reflects the simulated user's confidence in the correct state between every pair of variables (A->B, B->A, or no link), e.g. k=0.33 means flat prior for one third of variable pairs.

"The synthetic data set is created by generating a random directed acyclic graph and then sampling the graph with random edge weights for data sets of varying sizes. Each graph is generated with a sparsity of 0.3: each pair of variables has an edge between them with a probability of 0.3. Acyclicity is ensured by orienting all edges in the order the variables are defined, away from the first variable. The noise for each variable follows a zero-mean distribution which is randomly chosen from two options: either uniform distribution (-0.01, 0.01) or Gaussian with a standard deviation of 0.01. The reason for including both types of noise distributions is to create data sets which almost follow assumptions made by the algorithms while still breaking some of them. We tested creating data with different amounts of noise but that had no significant impact on the results. All of the algorithms we use in the experiments assume linearity but, additionally, PC-Stable and GES assume Gaussianity of noise and LiNGAM assumes non-Gaussianity."

4. Utilising expert knowledge and user interactions: here the authors suggest that not only user knowledge should be used for defining an optimal starting point (or graph) but also within a sort of interaction scheme between CSD and the user. What is the kind of procedure the authors have in mind and how is it used in the example? Is it a Bayesian-inspired approach with prior/CSD/posterior? More specifically, how is the "expert model" in (d) produced? Which CSD have been applied to generate (a) and to end up in (c)?

The process is Bayesian in nature, where the background "knowledge" manifests as known features in the graph and confidence in that knowledge and the user (in our case greedy search from neighbouring states) is presented with options for edits and can see how these edits would affect the model score. The "expert" model (d) was produced by two domain experts (Ivan Mammarella and Tuomo Nieminen). Graph (a) is produced by the PC-Stable algorithm. Assuming no background knowledge, interactive navigation of a Bayes-rational agent would result in graph (b). If the agent, again without any background knowledge, started from an empty graph instead of using an output of a CSD algorithm as the starting point, they would find the model represented by graph (c). Navigation without background knowledge is equal to greedily maximising the model likelihood (under the assumptions of linearity and Gaussianity).

5. Overfitting and concept drift: none of these two issues are developed in the section. I would recommend to either strengthen Section 5 with some experiment or simply move the observations in Sec5 to the conclusions. In summary, the paper addresses some crucial aspects regarding the application of CSD to scientific discovery. However, the interesting ideas presented are not developed in enough details. If the authors succeed in adequately expand their work then I would certainly recommend publication since the topic is of interest and can have potential impact in the forthcoming literature.

We can, if necessary, move the discussion about overfitting and concept drift to the conclusions.

Vehtari, A., Gelman, A. & Gabry, J. Practical Bayesian model evaluation using leave-one-out cross-validation and WAIC. *Stat Comput* **27**, 1413–1432 (2017). https://doi.org/10.1007/s11222-016-9696-4 OR https://arxiv.org/abs/1507.04544

---

## Author Response (AR1)

This document contains the referee comments to manuscript bg-2021-231: 'Comment on "Estimating causal networks in biosphere–atmosphere interaction with the PCMCI approach"' and author responses to these comments. The comments are presented as indented *cursive* text and we have numbered them when needed.

**Referee 1 Comments**

*The comment by Mäkelä et al. on the paper "Estimating causal networks in biosphere–atmosphere interaction with the PCMCI approach" by Krich et al. (including the reviewer) makes the point that the study should not take the outcome of a single causal discovery algorithm (here PCMCI) as an end result, but as a starting point and hypothesis for further study. They further illustrate on synthetic data how different prior expert knowledge affects such algorithms. The authors link to their recent workshop paper in the KDD 2021 conference (Melkas et al., 2021) which expands on the topic of "interactive" structure discovery.*

1. *Overall I deem this as a good and valid general point for any causal discovery analysis. However, I am not sure the commenting section is appropriate for this type of content since it does not specifically conduct an in-depth analysis of the paper to elaborate on how results would differ, but here it seems to mainly serve to advertise their workshop paper.*

We do agree that the paper does not conclusively solve the problem, and we have explained our motivation in the manuscript introduction. Our main interest is to highlight some pitfalls and caveats related to causal structure discovery (CSD) algorithms and to present these in a more general manner than in the workshop paper. Ideally, we would not have to be referring to our workshop paper, but since there is little other work in the nascent interactive CSD field available to illustrate our point, we refer the reader back to some of our work (which was motivated by wanting to avoid the exact pitfall that we point out here).

2. *The authors present three different aspects of what they call "user interaction": (1) starting from a user-guided initial state, (2) expert-interactions during the execution of the causal discovery algorithm, and (3) overfitting and concept drift. These three points are discussed with very minimal examples and a few bits are unclear from the text: Are all numerical analyses conducted with synthetic data? What's the setup? Can point (2) be elaborated on a bit, it is hard to understand how this interaction is meant here.*

We have modified the manuscript to better differentiate between each aspect and included a "workflow" description that should clarify the setting. In the manuscript, we use different CSD algorithms to produce our initial states (or empty graph) – this would provide a real user multiple starting points to choose from, e.g., what they regard as the best model in terms of both the model score and their background knowledge. In contrast, we provide the simulated user's background knowledge as likelihoods of graph structures (priors) and vary how much of this structure the "user" is aware of. The user (greedy search algorithm) then edits the graphs (starting from different initial states) using this knowledge. All presented numerical analyses use synthetic data, which enables us to know the "true model".

3. *To put these comments in context with the actual paper: Yes, in the scope of this paper (Krich et al.) no initial prior knowledge (other than the choice of variables and that the type of dependency is linear) was used. However, the resulting graph was discussed from an expert perspective. The problem of overfitting was addressed in so far that the hyper-parameter (pc_alpha) was optimized based on the Akaike Information Criterion, which is asymptotically equivalent to the cross-validation suggested in the comment. Indeed the paper can be viewed as a proof-of-concept and introduction to causality and underlying problems.*

We essentially agree with everything presented in the 3rd comment. We felt that it was important to demonstrate both these concepts to the wider audience and underline that measures should be taken

to avoid them. It is true that AIC is asymptotically equivalent to leave-one-out cross-validation (CV). However, we chose k-fold CV as our approach since it just looks at the test-set performance whereas AIC incorporates more subtle assumptions and that more parameters lead to a higher risk overfitting (and thus penalizes likelihood by the degrees of freedom). There are arguments both for and against each method and our deciding factor in favouring CV was simplicity. As a side note there is a nice comparison paper about these methods by Vehtari et al. (2016), the reference is at the end of this document.

4. *As a remark, including expert knowledge into causal discovery is an interesting and not quite trivial problem. For example, while it may be easy to code-up (PCMCI's software package Tigramite has an option to start from a user-given initial graph), the completeness ("maximal informativeness") of causal discovery algorithms under expert knowledge is an open problem, at least for more complex scenarios such as the presence of hidden variables.*

In the manuscript, we have not considered hidden or latent variables and we do agree with the comment. However, iterative user interaction still has a place as it may increase the user's understanding of the data and help in formulating beliefs about the potential data generating mechanisms. Additionally, more advanced methods could be developed to better identify and handle such situations.

**Referee 2 comments**

*The paper raises a warning with regard to the blind usage of causal structure discovery (CSD) algorithms and correctly suggests that CSD should be seen more as a useful guidance in the understanding and modeling of multivariate systems rather than as the final outcome of analysis. In particular, the authors identify four main weaknesses of CSD: different causality methods produce different results, the outcome depends on the initial graph, domain knowledge is not taken into account, overfitting and performance drops due to distribution shifts are usually ignored. All these concerns are of great interest and yet have not been sufficiently explored in the literature. For this reason, the paper could have an impact on how CSD are deployed for scientific discovery. However, the authors do not satisfactorily develop the ideas presented in the introduction or at least do not provide enough details. Even for a first contribution towards the interesting interplay between domain knowledge and CSD I think that the authors should make an extra effort and expand further the content of the paper. Find below more detailed comments per section.*

1. *Main text: what do you mean by "outcomes (models) of causal structure discovery (CSD) algorithms are, in many cases, interchangeable"? are you referring to the fact that different CSD may produce distinct causal graph over the same data? Perhaps explain or rephrase this sentence. How exactly does a greedy search over models help injecting previous knowledge into CSD?*

The offending sentence has been modified to: 'Different CSD algorithms may produce distinct outcomes (models) when operating on the same data. It is often difficult to identify the "correct" among these models, purely based on data.' The greedy search is made aware of (some) of the expert knowledge and is left to modify the initial graph (be it from a CSD algorithm or empty graph) by searching the neighbouring states (one edit away) and choosing the new state based on model score.

2. *Differences in CSD algorithms: the authors should provide a concrete example of a case in which different CSD produce different results (maybe taken from the literature). In its current form, section 2 is too generic and does not really add much information besides what is already stated in the introduction.*

We have added our example of this behaviour but also included a reference to a paper by Diego Colombo and Marloes H. Maathuis that demonstrates this behaviour among other things.

3. *The choice of initial state: this is one of the main parts of the paper where the authors describe their first experiment on how the initial graph affects the result. Interestingly, they show that increasing the level of prior knowledge the outcomes gradually converge. However, more details on how the experiment is performed should be given. What are the synthetic data used? Which are the four CSD methods? How is the parameter "k" (encoding the prior) used to generate initial states?*

We have modified the text and added the required information. Modifications have been made both to "Choice of initial state" -section as well as the main text, where we describe the synthetic data generation process. The CSD algorithms have been named both in text and images and a more detailed "k" descriptions has been added.

4. *Utilising expert knowledge and user interactions: here the authors suggest that not only user knowledge should be used for defining an optimal starting point (or graph) but also within a sort of interaction scheme between CSD and the user. What is the kind of procedure the authors have in mind and how is it used in the example? Is it a Bayesian-inspired approach with prior/CSD/posterior? More specifically, how is the "expert model" in (d) produced? Which CSD have been applied to generate (a) and to end up in (c)?*

We have added a workflow description to the manuscript as well as providing more details on the process itself. Information on different models etc. have also been appropriately added.

5. *Overfitting and concept drift: none of these two issues are developed in the section. I would recommend to either strengthen Section 5 with some experiment or simply move the observations in Sec5 to the conclusions. In summary, the paper addresses some crucial aspects regarding the application of CSD to scientific discovery. However, the interesting ideas presented are not developed in enough details. If the authors succeed in adequately expand their work then I would certainly recommend publication since the topic is of interest and can have potential impact in the forthcoming literature.*

We have now moved the discussion about overfitting and concept drift to the conclusions.

Vehtari, A., Gelman, A. & Gabry, J. Practical Bayesian model evaluation using leave-one-out cross-validation and WAIC. *Stat Comput* **27,** 1413–1432 (2017). https://doi.org/10.1007/s11222-016-9696-4 OR https://arxiv.org/abs/1507.04544

---

## Author Response (AR2)

This document contains editor comments to the revised version of bg-2021-231: 'Comment on "Estimating causal networks in biosphere–atmosphere interaction with the PCMCI approach"' and author responses to these comments. Editor comments are presented as indented text.

> **Editor comments**
>
> many thanks for your revised manuscript, and my apologies for getting back to you with a delay. I find that your manuscript has been improved given the suggestions by the reviewers. However, I think that the current manuscript contains too much jargon and unexplained/referenced acronyms, to make the content accecssible to the general readership of Biogeosciences. One, but not the only example for this is "The CSD methods we have used are PC-Stable with two significance levels 0.1 and 0.01, GES, and ICA-based LiNGAM.", where neither PC-Stable not ICA-based LiNGAM are explained. Please try to reformulate to make this accessible to non-specialist scientific readers.

We have now reworked parts of the manuscript for this end. In the beginning of main text we have added explanation about the PCMCI algorithm. Model score is now defined in the "numbered list" (workflow presentation). The algorithm acronyms are also now explicitly stated with references and we have added a paragraph detailing what are the fundamental differences between these approaches. Additionally Fig. 3 has been modified and panel captions removed.

> The revised version deals better with the conference paper issue and the remaining issue from Kirch et al. 2020, however, this also means that the title "Comment on:. ... " is not that appropriate anymore. I would suggest to use the technical note category of Biogeosciences instead, with a slightly modified title, and a slight alternation of the beginning of the Abstract. It is fine to start by stating that you provide a perspective of the Krich study.

The title has now been changed to "Technical note: incorporating expert domain knowledge into causal structure discovery workflows" and the beginning of the abstract modified. Some minor additional modifications have been made – the manuscript is now referred to as a "brief" or "note".